# Development of Visual Loop-Mediated Isothermal Amplification Assays for Foodborne Hepatitis A Virus

**DOI:** 10.3390/foods14060934

**Published:** 2025-03-10

**Authors:** Tongcan An, Mengyuan Song, Xiang Li, Yingjie Pan, Yong Zhao, Haiquan Liu

**Affiliations:** 1College of Food Science and Technology, Shanghai Ocean University, Shanghai 201306, China; qwdfghkm@126.com (T.A.); smyyykx@163.com (M.S.); yjpan@shou.edu.cn (Y.P.); 2Technical Center for Animal, Plant and Food Inspection and Quarantine of Shanghai Customs, Shanghai 201315, China; idealne@163.com; 3Shanghai Engineering Research Center of Aquatic-Product Processing & Preservation, Shanghai 201306, China; 4Laboratory of Quality & Safety Risk Assessment for Aquatic Products on Storage and Preservation (Shanghai), Ministry of Agriculture, Shanghai 201306, China; 5Engineering Research Center of Food Thermal-Processing Technology, Shanghai Ocean University, Shanghai 201306, China

**Keywords:** hepatitis A virus, visualization detection, lateral flow dipstick, colorimetric method

## Abstract

(1) Background: There are many cases of human disease caused by the hepatitis A virus contamination of aquatic products, so the development of the rapid detection of hepatitis A virus in aquatic products is crucial. (2) Methods: In this study, we developed three visual loop-mediated isothermal amplification methods for the rapid and intuitive detection of hepatitis A virus in aquatic products. New specific LAMP primers were designed for the HAV-specific VP1 protein shell. (1) HNB dye was added to the LAMP reaction system. After the reaction, the color of the reaction mixture changed from violet to sky blue, showing a positive result. (2) Cresol red dye was added to the LAMP reaction system, and a positive result was indicated by orange, while a negative result was indicated by purple. (3) By labeling FIP with biotin and LF with 6-FAM, the amplified product simultaneously contained biotin and 6-FAM, which bound to the anti-biotin antibody on the gold nanoparticles on the lateral flow dipstick (LFD). Subsequently, biotin was further combined with the anti-fam antibody on the T-line of the test strip to form a positive test result. (3) Results: The three visual LAMP methods were highly specific for HAV. The sensitivity of the visual assay was 2.59 × 10^0^ copies/μL. The positive detection ratio for 155 bivalve shellfish samples was 8.39%, which was the same as that for RT-qPCR. The three visual LAMP methods established in our work have better sensitivity than the international gold standard, and their operation is simple and requires less time. (4) Conclusions: The results can be obtained by eye color comparison and lateral flow dipsticks. Without the use of large-scale instrumentation, the sensitivity is the same as that of RT-qPCR. The test strips are lightweight, small in size, and easy to carry; they are suitable for emergency detection, on-site monitoring, field sampling, or remote farms and other non-laboratory environments for rapid identification.

## 1. Introduction

Hepatitis A virus (HAV) was first discovered by Feinstone [1] in 1973 in the feces of patients in the acute phase using immunoelectron microscopy. Hepatitis A virus (HAV) is a genus of hepatophilic viruses in the family of small RNA viruses [2]. HAV is spherical, with a diameter of about 27 nm, a molecular weight of about 7.5 kb, and no envelope. The capsid consists of 60 shell particles in a 20-plane stereo-symmetry [3]. In recent years, although the incidence of hepatitis A around the world has shown a downward trend, it has been found that HAV is still widespread in the external environment according to metagenomic studies [4]. It causes human infection mainly through the fecal–oral–digestive tract route, with foods such as raw foods, frozen foods, shellfish, and aquatic products acting as HAV vectors due to contamination [5,6], and there is now also literature indicating that baked goods such as sandwiches, salads, and breads are susceptible to contamination with hepatitis A virus [7]. During processing, transportation, storage, and distribution, they are susceptible to contamination and rapid multiplication, which can be harmful to humans. The incidence of HAV is significantly higher in areas where people consume raw food [8]. When HAV infects human liver cells, it replicates. Infected liver cells become inflamed and die under the attack of immune cells, eventually leading to impaired liver function in the body. The symptoms of hepatitis A include fatigue, loss of appetite, anorexia, and abnormal liver function. HAV is rarely fatal and can be controlled as long as the symptoms are detected in time, but, because of the long incubation period, the clinical manifestations range from asymptomatic infection to acute liver failure. Accidental infection with HAV occurs every year around the world, and a few people die due to untimely medical treatment [9]. Therefore, there is a need to develop rapid, sensitive, highly specific, and easy-to-use technologies for the detection of HAV in food.

Microbial culture methods [10], polymerase chain reaction (PCR) methods [11], and immunological detection [12] techniques can detect pathogenic microorganisms in food. However, these methods are limited by the need for laboratory instrumentation and operators, and they are unable to detect hepatitis A virus in a timely and effective way in remote areas with limited resources. With the rapid development of molecular diagnostic techniques, a number of isothermal amplification methods have been developed and applied to on-site detection. The loop-mediated amplification (LAMP) technique [13,14] is representative of isothermal techniques and has now been developed into a mature system. It uses four to six specially designed primers to ensure highly specific results, and, after heating in a simple water bath, it can produce about 10^3^ times more amplification products than the traditional PCR in a short time. It has the advantage of saving time and reducing costs. Due to its superior performance, the LAMP method has been shown to be a reliable and robust method for the detection and characterization of viral and microbial pathogens [15].

The purpose of this study was to create a rapid and easy technique to detect hepatitis A virus in food. In this study, new specific LAMP primers were designed for the hepatitis A virus-specific VP1 protein shell. A colorimetric LAMP and immunochromatographic method was developed to differentiate hepatitis A virus from other pathogenic bacteria, with the aim of applying this method to the on-site detection of hepatitis A virus in food.

## 2. Materials and Methods

### 2.1. Reagents and Materials

The genomic nucleic acids of *Listeria monocytogenes*, *E. coil*, *Vibrio parahaemolyticus*, GII norovirus, rotavirus, and sapovirus used in this experiment were kept in the laboratory to validate the specificity of the subsequent LAMP experiments.

Bst-DNA polymerase blockbuster enzyme (8 U/µL), MgSO_4_ (100 mM), and ThermoPol buffer (10×) were obtained from the Nanjing Novozymes Biotechnology Co., Ltd. (Nanjing, China). HNB dye (G1218) was purchased from the Shanghai Solarbio Biotechnology Co., Ltd. (Shanghai, China). Hepatitis A virus RNA standard (BNCC 369773) was obtained from the North China Biological Technology Co., Ltd. (Xinyang, China). Hepatitis A virus inactivation particles were purchased from the Tianjin KMD Technology Development Co., Ltd. (Tianjin, China). AgarD, 50×TAEbuffer, all primers, and plasmid standards were obtained from the Sangon Bioengineering (Shanghai) Co., Ltd. (Shanghai, China). The Tiangen Bacterial Genomic DNA Extraction Kit (DP302) was obtained from the Tiangen Biochemical Technology (Beijing) Co., Ltd. (Beijing, China). Highly pure dNTPs (10 mM) were obtained from the TransGen Biotech Co., Ltd. (Beijing, China). The RNApure Fast Tissue & Cell Kit (CW0599T) and RT-qPCR Mix (CW3371) were obtained from the Kangwei Century Biotech Co., Ltd. (Taizhou, China). WarmStart^®^ RTx Reverse Transcriptase (M0380S) was obtained from the NEB Biological Technology (Beijing) Co., Ltd. (Beijing, China). Lateral flow test strips were obtained from the Wuhan Genenode Biotechnology Co., Ltd. (Wuhan, China). Cresol red dye and PEG8000 were obtained from the Shanghai McLean Biochemical Technology Co., Ltd. (Shanghai, China). All solutions were prepared with enzyme-free water.

### 2.2. Bacterial Recovery Culture and DNA Extraction

Bacterial pure culture [16,17]: *Listeria monocytogenes*, *E. coli*, and *Vibrio parahaemolyticus* were taken out from the −80 °C refrigerator. This step of bacterial resuscitation is of fundamental importance. According to a 1:100 ratio, 50 μL of the bacterial solution was sequentially added to 5 mL of the LB culture solution. The mixture was then placed in a shaker operating at 220 r/min and incubated at 37 °C overnight for 12 h. After the initial 12 h incubation, the plate of the resuscitated bacterial solution was taken to perform line drawing. The incubation was continued for another 12–18 h. After checking that there were no stray bacteria, single colonies were picked and transferred to 10 mL of LB culture solution once again and then cultured overnight in the shaker for subsequent experimental operations.

Bacterial DNA extraction: 1 mL of three types of bacteria after recovery (OD600 = 1.0 or so) was taken. The genomic DNA of *Listeria monocytogenes*, *E. coli*, and *Vibrio parahaemolyticus* was extracted according to the instructions of the Tiangen Bacterial Genomic DNA Extraction Kit (DP302), and the concentration and quality of the DNA was checked by an ultra-micro UV spectrophotometer and stored in the refrigerator at −20 °C for further use.

### 2.3. Bivalve Molluscan Shellfish (BMS) Samples—RNA Extraction

The surfaces of the fresh bivalve molluscan shellfish samples were washed, and the water on the surface of the shell was naturally drained so that there was no flowing water on the surface. The larger and more active bivalve shell samples were weighed. Then, they were placed in a foam tank filled with ice for the next step. A mask was worn, and an ice bag was placed on a clean and flat operating table covered with tin paper to maintain a low-temperature environment. The shell was cut with a surgical razor, and the digestive gland was collected into small pieces. Then, 2 g of the digestive gland from each sample was collected into a 1.5 mL centrifuge tube and weighed, and the data were recorded. The gloves and blades were replaced for each sample. According to the method of virus enrichment in ISO 15216–2:2019 [18], some modifications were made [19,20]. After the digestive gland was extracted, 500 μL normal saline was added, and it was ground at a low temperature. Then, 400 μL of the supernatant was taken and mixed evenly with a 40% PEG 8000 solution (prepared with enzyme-free water) to allow the final concentration of PEG 8000 to reach 10%. The mixture was then centrifuged at 12,000× *g* rpm for 30 min at 4 °C. Excess supernatant was removed, and the final volume of the mixture was 200 μL. RNA was extracted immediately according to the instructions of the RNApure Fast Tissue & Cell Kit (CW0599T) from the Kangwei Century Biotech Co., Ltd.

### 2.4. Preparation of Artificially Contaminated Samples and RNA Extraction

The RT-LAMP-LFD experiment and RT-qPCR experiment were carried out by using artificially contaminated simulated samples to evaluate the feasibility of the RNA extraction method. The results were judged using a 2% electrophoresis gel imaging system, lateral flow test strips, and RT-qPCR data. Twenty *Scapharca subcrenata* were subjected to water purification treatment for more than 24 h [21] to keep them HAV-negative. Then, 20 digestive glands of *Scapharca subcrenata* were taken; 10 of them were combined with 10 μL of hepatitis A virus-attenuated vaccine to create artificially contaminated samples [22], and the other 10 were combined with 10 μL of non-enzymatic water as the negative control group. Both were placed on ice for 20 min, so that the virus was fully adsorbed in the sample. Then, we performed the RNA extraction as for the BMS samples. We calculated the detection rate to evaluate whether the extraction method was feasible.

### 2.5. LAMP Primer Design

Through the NCBI, the highly conserved sequence of the hepatitis A virus-specific shell protein VP1 gene fragment (GenBank: AY334041.1) was identified using the BLAST program (BLASTN). All LAMP primers were designed using the PrimerExplorer V5 online software (http://primerexplorer.jp/lampv5e/index.html (accessed on 7 November 2023)), and we screened out three sets of primers for the hepatitis A virus-specific fragment, named groups 1, 14, and 97 (Table 1). Primer screening was performed under the same amplification conditions and analyzed by AGE.

### 2.6. LAMP Reaction System Optimization

In order to obtain the optimal system for the detection of hepatitis A virus by LAMP, we used three sets of primers to optimize the 25 μL LAMP amplification system, focusing on the Bst DNA polymerase concentration (0 U/μL, 0.16 U/μL, 0.32 U/μL, 0.48 U/μL, 0.64 U/μL), Mg^2+^ concentration (2 mM, 4 mM, 6 mM, 8 mM, 10 mM, 12 mM, 14 mM), dNTP Mix concentration (0.6 mM, 0.8 mM, 1.0 mM, 1.2 mM, 1.4 mM, 1.6 mM, 1.8 mM, 2.0 mM), 10 mM internal/external primer addition (4.0 μL:0.5 μL, 4.0 μL:0.25 μL, 3.0 μL:0.5 μL, 2.0 μL:0.5 μL, 1.0 μL:0.5 μL, 1.0 μL:1.0 μL, 1.0 μL:2.0 μL), reaction temperature (60 °C, 61 °C, 62 °C, 63 °C, 64 °C, 65 °C), loop primer concentration (0.25 mM, 0.50 mM, 0.75 mM, 1.00 mM, each primer), reaction time (30 min, 35 min, 40 min, 45 min, 50 min, 55 min, 60 min, 65 min), and WarmStart^®^ RTx Reverse Transcriptase addition (0.5 μL, 0.75 μL, 1 μL, 1.25 μL) [23,24]. We used 1 μL template for all experiments. All optimization experiments had a negative control group. Through the 2% agarose gel electrophoresis imaging test results, the best primer group was selected by considering the brightness of the selected bands and the amount of reagent added in the LAMP reaction. The specificity was also tested.

### 2.7. Visual LAMP Reaction System

A visual LAMP reaction system was established using the optimized reverse transcription LAMP system.

HNB-LAMP reaction system: The combination of HNB and Mg^2+^ rendered the initial color of the LAMP reaction system violet. As the reaction progresses, Mg^2+^ reacts with P_2_O_7_^2−^ to yield a white precipitate, HNB loses Mg^2+^, and the system’s color changes to sky blue, while the unreacted system still remains violet [25].

Cresol red–LAMP reaction system: Cresol red is an acid–base indicator. During the LAMP reaction, H^+^ will be produced when each dNTPS is consumed, resulting in a gradual decrease in the pH of the reaction solution. In the 25 μL LAMP reaction system, the pH of the reaction solution was reduced by about 2 after the complete reaction [26,27]. In the reaction with the addition of cresol red, the positive result was orange and the negative result was purple.

LFD-LAMP reaction system: Biotin was used to label the LAMP-specific primer FIP, and 6^^^carboxyl fluorescein (6-FAM) was used to label the LAMP-specific loop primer LF. Both of them specifically bound to the target fragment, and the LAMP product was detected by LFD. The color development of C- and T-lines indicates a positive result, and the color development of a C-line alone indicates a negative result [28]. The color of the T-line and the color of the C-line indicate that the test strip is faulty and needs to be resampled.

### 2.8. Sensitivity Experiment

Different concentrations of 2.59 × 10^4^ copies/μL, 2.59 × 10^3^ copies/μL, 2.59 × 10^2^ copies/μL, 2.59 × 10^1^ copies/μL, and 2.59 × 10^0^ copies/μL RNA were obtained by diluting the 2.59 × 10^5^ copies/μL hepatitis A virus standard as a 1 μL template to evaluate the detection limit. The limit of detection (LOD) was, respectively, evaluated by LAMP-AGE, colorimetric LAMP, and LAMP-LFD.

### 2.9. LAMP-LFD Stability and Repeatability Experiments

The minimum detection limit of the established RT-LAMP-LFD method was selected as the template. The experiment was conducted by three different operators. The 2.59 × 10^1^ copies/μL concentration of the hepatitis A virus RNA standard was selected as the positive sample, and non-enzymatic water was taken as the negative control. The results were detected by a lateral flow test strip and 2% AGE. We observed whether there was a strip in the gel map and whether the C- and T-line of the test strip could still be clearly displayed, and we counted the test results.

### 2.10. Visual LAMP Detection of Actual Samples

A total of 155 RNA samples were detected by the HNB-RT-LAMP, cresol red–RT-LAMP, and RT-LAMP-LFD systems. The detection rate of positive samples was calculated.

### 2.11. RT-qPCR Standard Curve

The hepatitis A virus RNA was diluted to the following concentrations to obtain the standard curve of the RT-qPCR: 5.2 × 10^4^ copies/μL, 5.2 × 10^3^ copies/μL, 5.2 × 10^2^ copies/μL, 5.2 × 10^1^ copies/μL, and 5.2 × 10^0^ copies/μL. The simple linear regression equation of the relationship between the concentration of hepatitis A virus RNA and Ct was obtained. The specific experimental operations were as follows: 25 μL of the final system was configured in the biosafety cabinet, including the HAV68 primer 10 mM 1 μL, HAV240 primer 10 mM 1 μL, hybridization probe HAV150 10 mM 0.5 μL, reaction buffer 5 μL, enzyme mixture 1 μL, RNA 5 μL, and deionized water, which was supplemented to 25 μL. We used the Yena Tower^3.0^ fluorescence quantitative PCR instrument and the ISO standard program, which was slightly modified according to the following RT-qPCR amplification program: 55 °C reverse transcription for 30 min; pre-denaturation at 95 °C for 3 min; after 45 cycles of denaturation at 95 °C for 15 s and annealing at 61 °C for 30 s, the fluorescence was collected without ROX correction. The blank control group was treated with non-enzymatic water, and the negative control group was treated with norovirus RNA. After the reaction, the amplification results were judged by the fluorescence signal.

### 2.12. RT-qPCR Detection of Actual Samples

According to the previous procedure, 155 samples were detected and 5 μL RNA was loaded. The blank control group was treated with non-enzymatic water, and the negative control group was treated with norovirus RNA. After the reaction, the amplification results were judged by the fluorescence signal. The detection rate of positive samples was calculated from the statistical results.

## 3. Results and Discussion

### 3.1. Optimization of LAMP Reaction

The LAMP reaction was analyzed by 2% AGE in order to obtain the optimal LAMP amplification system (Figure 1). Considering the brightness of the amplification product on the gel, the cost of the reagent, and other factors, the optimal LAMP system consists of Bst DNA polymerase at concentrations of 0.48 U/μL, 0.32 U/μL, and 0.32 U/μL. The final concentration of Mg^2+^ is 8 mM, 8 mM, and 6 mM. The dNTP mix concentration is 1.6 mM, 1.8 mM, and 1.4 mM. The internal and external primer addition amounts are 4.0 μL:0.25 μL, 4.0 μL:0.5 μL, and 3.0 μL:0.5 μL. The reaction temperatures tested were 62 °C, 61 °C, and 61 °C. The concentrations of each loop primer are 0.25 mM, 0.50 mM, and 0.50 mM, and the reaction time is 30 min, 30 min, and 35 min.

### 3.2. Primer Screening

Online access to the NUPACK website (https://www.nupack.org/) [29] was used to simulate the secondary structure formation status of each of the three sets of designed LAMP primers themselves at 63 °C amplification. The results are shown in Figure 2. Figure 2A is the equilibrium concentration diagram, which was used to simulate the steady states of the three groups of primers 1, 14, and 93 after a period of time in the LAMP reaction (there may be stable complementarity between chains, self-complementation, or a chain in the form of a single chain). From Figure 2A, it can be seen that all three sets of primers have no chain-to-chain interaction, and the concentrations of the primers are almost unchanged. This indicates that, during the LAMP reaction, there will be no mismatch between the primers in the system, resulting in a decrease in the primer concentration in the system and affecting the sensitivity of the reaction, and no secondary structure between the primers will be generated. Figure 2B is the overall pairing coefficient diagram of the system chain, and Figure 2C is the pairing probability heat map for a single base pair. Both of them show the base pair position and expected probability of each primer’s secondary structure self-complementation in the LAMP system. The redder the color, the greater the self-complementation probability, and the maximum is 100%. From Figure 2B, it can be seen that 1-FIP, 1-BIP, 1-F3, and 1-LB in primer group 1; 14-FIP, 14-BIP, 14-F3, and 14-LB in primer group 14; and 97-BIP, 97-F3, 97-B3, and 97-LB in primer group 97 did not show the possibility of self-complementation, while 1-B3 and 14-B3 had two matches, and 14-LF and 97-FIP had four matches. From the chain minimum free energy (MFE) visual pattern in Figure 2D, the base pair sites and probabilities of the above four primers’ self-matched secondary structures can be seen. An analysis of the diagram shows that 1-B3 and 14-B3 may have a secondary structure of two base pairs, but the two base pairs themselves are extremely unstable and can easily untie, so they have no major influence on the reaction. Meanwhile, 14-LF and 97-FIP have the possibility of a four-base-pair secondary structure, and the matching is more stable. However, from the heat map analysis, the 14-LF primer’s pairing point shows a light yellow–green color, while the 97-FIP primer’s pairing point shows a light orange color, and their minimum free energy absolute values are less than 3.5 kcal/mol, indicating that the two primers are not likely to be self-complementary. Therefore, the above three groups of primers can be used in experiments in theory, but, based on their actual performance, we selected 14 primers as the optimal primer group to conduct the following experiment.

### 3.3. Establishment of Visual LAMP Reaction System

The RT-LAMP system was as follows: 2.5 μL 10 × isothermal amplification buffer, 4.5 μL 10 mM dNTPs, 1.5 μL 10 mM MgSO_4_, 4 μL 10 μM FIP/BIP, 0.5 μL 10 μM F3/B3, 1 μL Bst DNA polymerase, 1 μL RNA, 0.5 μL 10 μM LF/LB, and WarmStart^®^ RTx Reverse Transcriptase 0.5 μL. We added non-enzymatic water to 25 μL, creating a blank control without any template RNA. The reaction mixture was incubated at 61 °C for 30 min and then heated at 80 °C for 5 min to terminate the reaction.

For the visual HNB-LAMP system and cresol red–LAMP system, we only need to add 1 μL of 25 × HNB dye solution (diluted with non-enzymatic water to ensure that the final concentration of the system is 1×) and 1 μL of 2.6 mol/L cresol red dye solution, respectively, and non-enzymatic water was supplemented to 25 μL. After the system was prepared, 20 μL of liquid paraffin was added to prevent the dye from transpiring to the lid of the PCR single tube during the reaction. In the reaction with the addition of HNB, a positive result was indicated by light blue, and a negative result was indicated by violet. In the reaction with the addition of cresol red, a positive result was indicated by red–orange and a negative result by purple.

A visual RT-LAMP-LFD system was constructed. The LAMP reaction product was mixed with non-enzymatic water to a total volume of 100 μL to obtain a test strip sample test solution. The test solution was sucked and added to the sample pad of the lateral flow test strip for LAMP-LFD detection. It was left at room temperature for 5 min to wait for the result. Both colored C- and T-lines indicate a positive result, and a colored C-line indicates a negative result. A colored T-line and uncolored C-line indicate that the test strip is faulty and needs to be retested.

### 3.4. Sensitivity Experiment

By diluting a series of 2.59 × 10^5^ copies/μL hepatitis A virus RNA standards, 2.59 × 10^4^ copies/μL, 2.59 × 10^3^ copies/μL, 2.59 × 10^2^ copies/μL, 2.59 × 10^1^ copies/μL, and 2.59 × 10^0^ copies/μL were obtained. At each dilution gradient, 1 μL of RNA template was taken for the RT-LAMP sensitivity test. The reaction mixture was incubated at 61 °C for 30 min and terminated at 80 °C for 5 min. The product was detected by a lateral flow test strip and 2% AGE. See Figure 3, where ‘N’ represents the negative control group, and 1–6 in turn indicate the addition of 1 μL RNA template at the concentrations of 2.59 × 10^0^ copies/μL, 2.59 × 10^1^ copies/μL, 2.59 × 10^2^ copies/μL, 2.59 × 10^3^ copies/μL, 2.59 × 10^4^ copies/μL, and 2.59 × 10^5^ copies/μL. The result showed that the minimum detection limit of the three optimized visual LAMP methods for hepatitis A virus was 2.59 × 10^0^ copies/μL.

### 3.5. LAMP Specificity Experiment

The optimized LAMP method was used to detect seven types of DNA, namely sapovirus, rotavirus, norovirus, hepatitis A virus, *E. coli*, *Listeria monocytogenes*, and *Vibrio parahaemolyticus*. The specificity of the primers was verified, and the results were judged using a 2% electrophoresis gel imaging system. As shown in Figure 4, only the specific bands of the hepatitis A virus genome were amplified, and the corresponding bands of the other genomic DNA used for verification were only the primer dimers below, indicating that they were not specifically amplified and that primer No. 14 exhibited strong specificity.

### 3.6. LAMP-LFD Stability and Repeatability Experiments

The results are shown in Figure 5. The detection rate regarding the positive results detected by the three parallel tests of the three experimenters was 100%. The reaction products were observed by 2% AGE, and the C- and T-line reaction results could be clearly observed on the test strip, without false positive test results. This shows that the RT-LAMP-LFD method described in this paper is repeatable and stable.

### 3.7. Standard Curve of RT-qPCR

The standard curve y = 42.71 − 3.52x (R^2^ = 0.98043 > 0.98000) and the PCR amplification efficiency was 93% (Figure 6). The detection limit was estimated to be 5.2 × 10^0^ copies/μL.

### 3.8. Artificial Simulation Pollution Experiment

Twenty artificially contaminated samples were used as templates, with 1 μL template for the RT-LAMP-LFD experiment, 5 μL template for the RT-qPCR experiment, and a blank control with non-enzymatic water. The results were judged using 2% AGE and the RT-qPCR data. As shown in Figure 7, the artificially contaminated samples (1–10) showed positive results on the 2% AEG, which were consistent with the results of the RT-qPCR. The detection rates for positive samples using RT-LAMP-LFD and RT-qPCR were 100%, and the negative control samples were not detected. This shows that the extraction method for hepatitis A virus established in this paper is accurate and reliable and can be used to extract hepatitis A virus RNA in actual samples.

### 3.9. Visual LAMP Detection of Actual Samples

The RNA extracted from the digestive glands of 155 bivalve shellfish aquatic products was used as a template, and the visual HNB-RT-LAMP, cresol red–RT-LAMP, and RT-LAMP-LFD methods were used for detection. All experiments were repeated three times to ensure the repeatability of the experiment.

The results showed that all three methods detected positive samples, and the number of positive samples detected was consistent with the corresponding number, maintaining a high degree of accuracy. Among the 155 bivalve shellfish aquatic products, a total of 13 samples showed positive results, and the other samples were negative. The positive samples were numbered 10, 13, 26, 72, 96, 100, 102, 107, 114, 115, 126, 127, and 133 (Figure 8). The positive sample detection rate of the statistical visual LAMP method was 8.39%.

### 3.10. RT-qPCR Detection of Actual Samples

The extracted RNA from the digestive glands of 155 bivalve shellfish was used as a template and detected by RT-qPCR. Negative samples had no Ct values, and positive samples had Ct values. All experiments were repeated three times to ensure repeatability. The results showed that the specific numbers of positive samples were 10, 13, 26, 72, 96, 100, 102, 107, 114, 115, 126, 127, and 133. The detection rate for positive samples with RT-qPCR was 8.39%. The results of the fluorescence quantitative RT-PCR were consistent with the results of the visual LAMP method established by us.

## 4. Conclusions

In this study, colorimetric LAMP and LAMP-LFD were developed for the rapid detection of HAV. A new specific LAMP primer was designed for the specific VP1 protein shell of hepatitis A virus, and a set of loop primers (LF/LB) was designed to improve the reaction speed and sensitivity of HAV detection. These analytical methods can distinguish HAV from other viruses, and their analytical sensitivity is consistent with the results of AGE, with a detection limit of 2.59 × 10^0^ copies/μL. It is expected that these visual methods can be used for real food samples. The detection rate for positive samples from actual bivalve shellfish samples with the visual LAMP method was 8.39%, which is consistent with the detection rate of the RT-qPCR method. The results showed that the sensitivity of the HNB-RT-LAMP, cresol red–RT-LAMP, and RT-LAMP-LFD methods established in this work for the detection of hepatitis A virus was equivalent to or even better than that of RT-qPCR, and the specificity and sensitivity were high. The whole detection process only took 30–35 min. Moreover, this technology does not require specialized laboratory equipment, making it faster and easier to operate. It provides a new development direction for the diagnosis and prevention of hepatitis A virus, and it is expected to become a routine and simple detection method, being especially suitable for grassroots and on-site detection applications.

## Figures and Tables

**Figure 1 foods-14-00934-f001:**
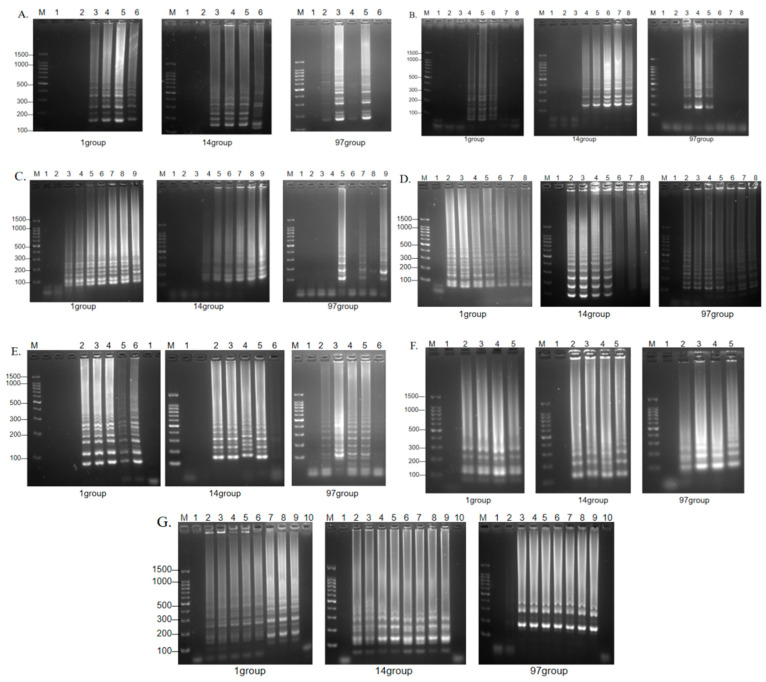
Optimization of LAMP reaction. (**A**) Bst DNA polymerase. (**B**) Final concentration of Mg^2+^. (**C**) dNTP Mix concentration. (**D**) 10 mM internal and external primer addition amounts. (**E**) Reaction temperature. (**F**) Loop primer concentration. (**G**) Reaction time. M: DNA 100 marker; 1, 10: negative control.

**Figure 2 foods-14-00934-f002:**
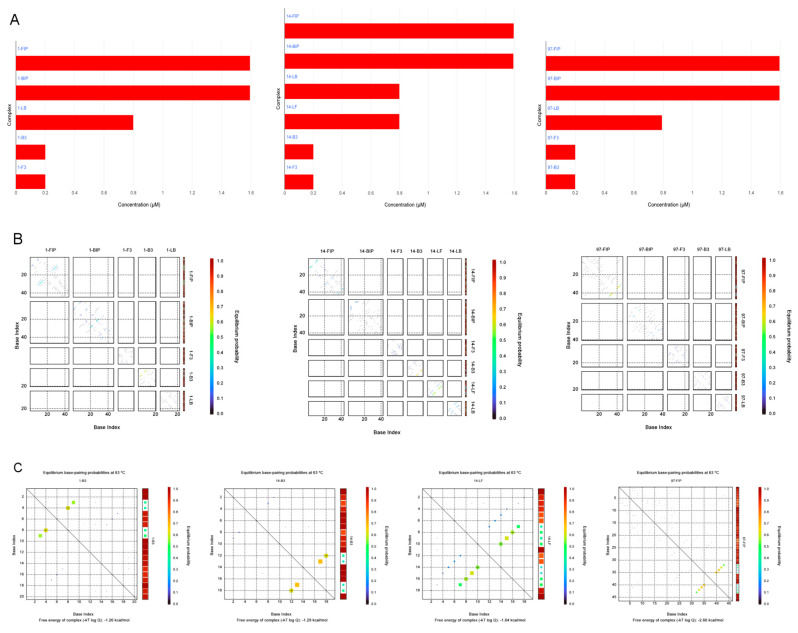
Flow chart of the secondary structures of the 3 sets of primers during the reaction simulated by the NUPACK software. (**A**) Reaction system equilibrium concentration. (**B**) Ensemble pair fractions of chains. (**C**) Pair probabilities heat map. From left to right are 1-B3, 14-B3, 14-LF, 97-FIP. (**D**) Minimum free energy visualization pattern. From left to right are 1-B3, 14-B3, 14-LF, 97-FIP.

**Figure 3 foods-14-00934-f003:**
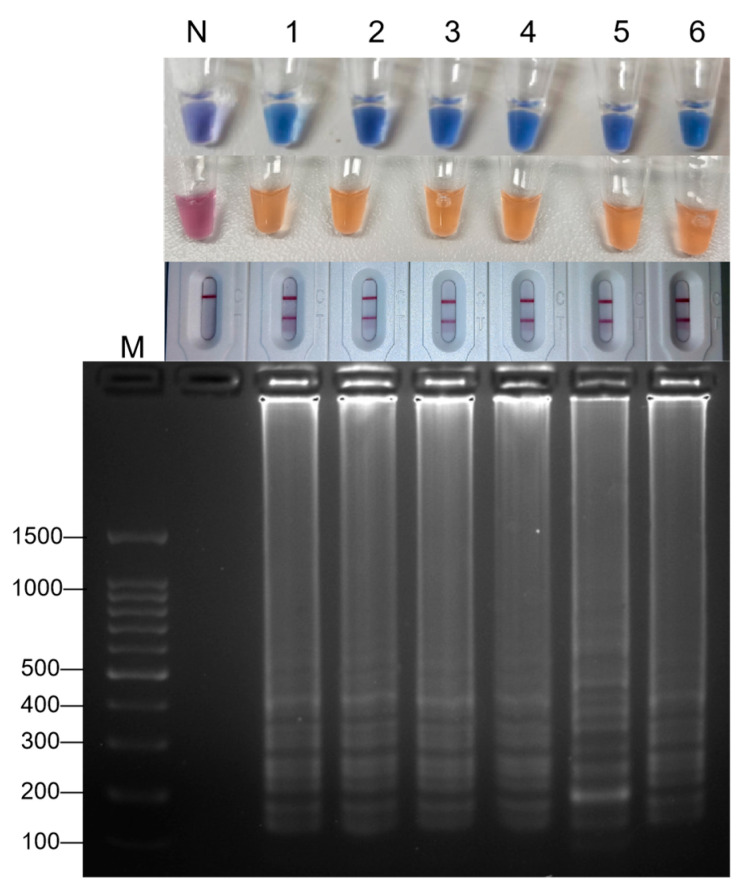
The results of the LAMP reaction under different concentrations of RNA. M: DNA 100 marker; N: negative control; 1–6: 1 μL RNA template concentrations of 2.59 × 10^0^, 2.59 × 10^1^, 2.59 × 10^2^, 2.59 × 10^3^, 2.59 × 10^4^, 2.59 × 10^5^ copies/μL.

**Figure 4 foods-14-00934-f004:**
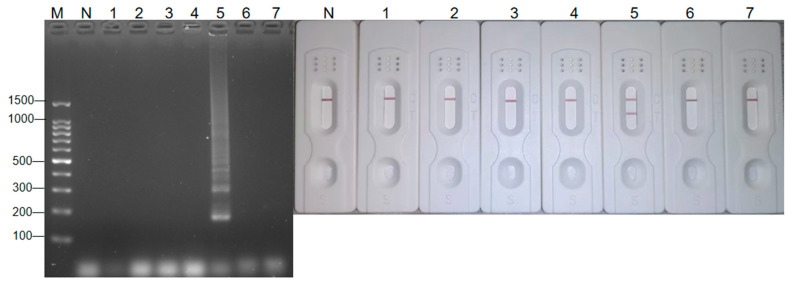
Specificity test results of the method. M: DNA 100 marker; N: control group; 1–7: sapovirus, rotavirus, norovirus, *E. coli*, HAV, *Listeria monocytogenes*, *Vibrio parahaemolyticus*.

**Figure 5 foods-14-00934-f005:**
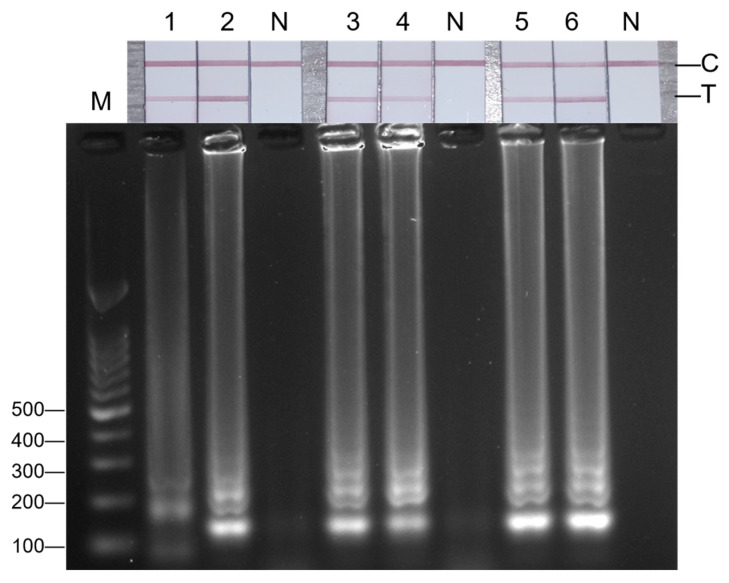
Repeated test of LAMP-LFD. M: DNA 100 marker; N: negative control.

**Figure 6 foods-14-00934-f006:**
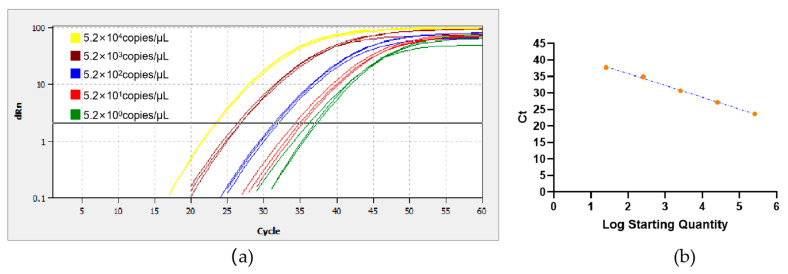
RT-qPCR (**a**) amplification plot and (**b**) standard curve diagram.

**Figure 7 foods-14-00934-f007:**
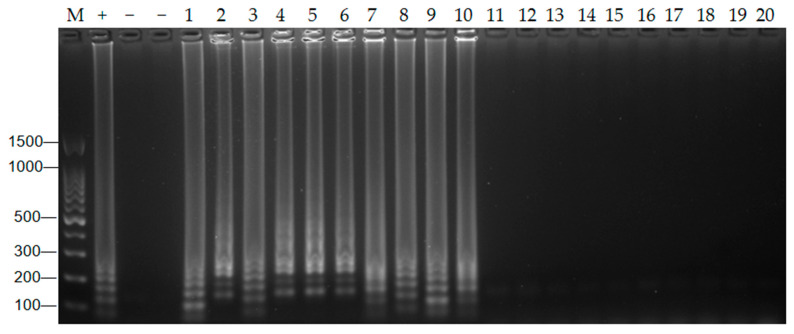
Results of artificial pollution simulation experiments. M: DNA 100 marker; +: positive control; −: negative control.

**Figure 8 foods-14-00934-f008:**
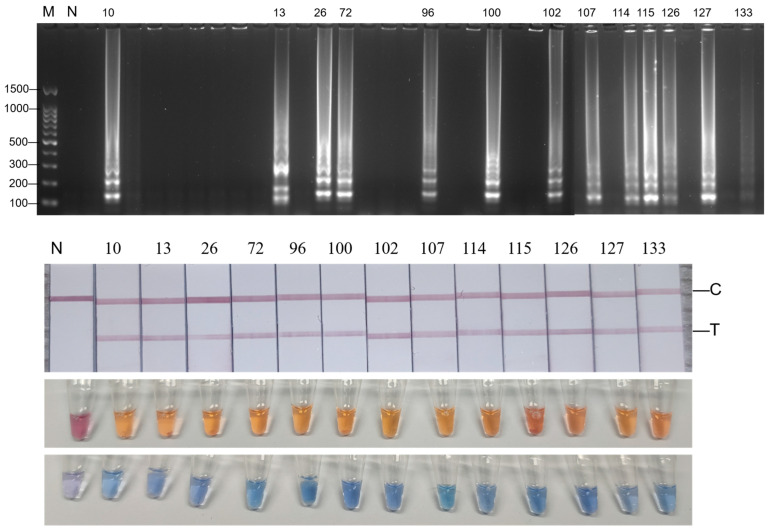
Results of three visual LAMP detection methods used on actual samples. M: DNA 100 marker; N: negative control.

**Table 1 foods-14-00934-t001:** The primers used in this study.

Primer Name		Sequence (5′-3′)
1	FIP	TCCCTCATGGTTGTTATGCCGTTTCAACAACAGTTTCTACAGAG
	BIP	CCTAAAAGGGAAAGCCAATAGGGGATCCTCAATTGTTGTAATAGCT
	F3	GAGACGATTCAGGGGGTT
	B3	TCAGGCACTTTCTTTGCTAA
	LB	GATGGATGTTTCAGGAGTGCAG
14	FIP	Biotin-TCCCTTTTAGGTCCCTCATGGTAGTTTCTACAGAGCAGAATGT
	BIP	AGCCAATAGGGGGAAGATGGATTGCTAAAACTGGATCCTCAA
	F3	CAGGGGGTTTTTCAACAAC
	B3	GGAAATGTCTCAGGCACTT
	LB	AGTGCAGGCACCTGTGG
	LF	6-FAM-GCCGATCTGAGGATCAGGA
97	FIP	CCAATGGTCCTCTATACAACTGAAACTCCTCATGGTTTACCATCAA
	BIP	GTGGATGGTATGGCCTGGTTATAGACAAAGCTGACTCCTT
	F3	CCAATAACTTTGTCTTCAACTTCT
	B3	TGAATCTAACAGCTCCAAGG
	LB	CTTGCTGTCGACACCCCT
HAV68		TCACCGCCGTTTGCCTAG
HAV240		GGAGAGCCCTGGAAGAAAG
HAV150		6-FAM-CCTGAACCTGCAGGAATTAA-MGB

## Data Availability

The original contributions presented in this study are included in the article. Further inquiries can be directed to the corresponding author.

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
