# Peer review of "Development of Visual Loop-Mediated Isothermal Amplification Assays for Foodborne Hepatitis A Virus"

_foods, 2025, doi:10.3390/foods14060934_

Round 1
Reviewer 1 Report
Comments and Suggestions for Authors
This study optimizes the LAMP reaction for hepatitis A virus detection, refining reaction conditions, primer selection, and visualization methods to enhance specificity, sensitivity, and efficiency, with results comparable to RT-qPCR, making it suitable for rapid, on-site diagnostics.
The study tested 155 bivalve shellfish samples, which may not be sufficient to fully validate the method’s reliability across diverse real-world conditions. A larger sample set would strengthen the findings. Environmental and food samples may contain inhibitors that affect nucleic acid amplification, but the study does not detail how the method performs in the presence of common inhibitors, maybe these infos could be added.
Some minor suggestions:
careful at the spaces left between words and commas. example: "system(Figure 1).Considering" - "system (Figure 1). Considering"; "diagram , which" "diagram, which"; "From the analysis of the diagram, it can be seen that" - "Analysis of the diagram shows that" (more concise);
Also, revise some phrases for clarity. Some of them are really shorttened but do not always reflect the meaning. examples:
The best system of the three sets of primers is Bst DNA polymerase 0.48 U/μL, 0.32 U/μL, 0.32 U/μL." coulb be rephrased into: "The optimal primer system consists of Bst DNA polymerase at concentrations of 0.48 U/μL, 0.32 U/μL, and 0.32 U/μL." - I think it sound better
or "The reaction temperature is 62 °C, 61 °C, 61 °C." could be "The reaction temperatures tested were 62 °C and 61 °C."
"indicating that the No.14 primer had strong specificity." could be rephrased into "indicating that primer No.14 exhibited strong specificity."
Also, the phrase "Moreover, this technology does not require special laboratory conditions of instruments and equipment, and is faster, easier to operate."could be "Moreover, this technology does not require specialized laboratory equipment, making it faster and easier to operate."
The article is good and after minor revisions it deserves to be published.
Author Response
Response to Reviewer 1 Comments
Thank you for your reply concerning our manuscript entitled “Development of Visual Loop-Mediated Isothermal Amplification Assays for foodborne Hepatitis A Virus” (Manuscript ID: foods-3473510). Those comments are all valuable and helpful for revising and improving our paper. We have revised our manuscript according to the comments and have highlighted in yellow the revised portion in the paper. Our point-by-point responses to the comments are given below; all the page and line numbers refer to those in the revised manuscript.
Thanks very much for your attention to our manuscript. If you have any questions about this paper, please don’t hesitate to let me know.
Point-by-point response to Comments and Suggestions for Authors:
Comments 1: [(1) The study tested 155 bivalve shellfish samples, which may not be sufficient to fully validate the method’ s reliability across diverse real-world conditions. A larger sample set would strengthen the findings. (2) Environmental and food samples may contain inhibitors that affect nucleic acid amplification, but the study does not detail how the method performs in the presence of common inhibitors, maybe these infos could be added.]
Response 1: [Thank you for your insightful comment and kind suggestion.
(1) We followed the sampling principle and collected enough samples from Wholesale market and farm product trade market, Shanghai, China. The samples covered the market of each district in the Shanghai. The above all samples were collected in July, August and September of 2024, which are the months with high hepatitis A incidence, which can investigate the HAV pollution of aquatic products in Shanghai from the side. The digestive glands are 2 g each portion, and we need two portions of digestive glands for each sample, one for the experiment, and one to save.
Finally, small samples without commodity value were removed. Combined with our test methods and reported literatures, we think these samples can represent the situation in Shanghai in these three months and support the establishment of test methods.
(2)First, we used the RNApure Fast Tissue&Cell Kit (CW0599T) to extract the RNA. This product adopts silica gel column purification technology, which can rapidly extract total RNA from animal tissues and cells without using toxic and inhibitors that affect nucleic acid amplification (Generally affects enzyme activity), such as β-mercaptoethanol and phenol/chloroform in the extraction process. The RNA obtained was of high yield, good integrity and free from contamination by other impurities. Second, It is reported in the literature that the concentrations of inhibitors that cause significant inhibition of LAMP reaction are close to the concentrations found in the real world. But it is likely that real samples are diluted (about 2 - 10 fold) prior to addition to LAMP reactions thereby lowering their concentrations to levels that do not inhibit LAMP, and that inhibitors may interact with other molecules in the sample thereby preventing them from inhibiting the components of the LAMP reaction, and lamp's buffer also has some resistance to inhibition. This paper mainly establishes qualitative detection. Combined with your suggestions, when we do subsequent quantitative research, common inhibitors do affect nucleic acid amplification, and we will fully consider the Suggestions of inhibitors in subsequent quantitative experiments.]
Comments 2: [ Some minor suggestions:
careful at the spaces left between words and commas. example: "system(Figure 1).Considering" - "system (Figure 1). Considering"; "diagram , which" "diagram, which"; "From the analysis of the diagram, it can be seen that" - "Analysis of the diagram shows that" (more concise);
Also, revise some phrases for clarity. Some of them are really shorttened but do not always reflect the meaning. examples:
The best system of the three sets of primers is Bst DNA polymerase 0.48 U/μL, 0.32 U/μL, 0.32 U/μL." could be rephrased into: "The optimal primer system consists of Bst DNA polymerase at concentrations of 0.48 U/μL, 0.32 U/μL, and 0.32 U/μL." - I think it sound better
or "The reaction temperature is 62 °C, 61 °C, 61 °C." could be "The reaction temperatures tested were 62 °C and 61 °C."
"indicating that the No.14 primer had strong specificity." could be rephrased into "indicating that primer No.14 exhibited strong specificity."
Also, the phrase "Moreover, this technology does not require special laboratory conditions of instruments and equipment, and is faster, easier to operate."could be "Moreover, this technology does not require specialized laboratory equipment, making it faster and easier to operate."]
Response 2: [Thank you for your valuable feedback and pointing this out! Recommendations on the proposed spaces left between words and commas, clarity of the sentence. We have all rephrased ! We appreciate your attention to detail and your help in enhancing the quality of the manuscript. Now is much smoother of the expression. Those changes can be found :
Line 241: “system (Figure 1).”
Line 242-243: “the optimal LAMP system consists of Bst DNA polymerase at concentrations of 0.48 U/μL, 0.32 U/μL, and 0.32 U/μL.”
Line 246-247: “The reaction temperature tested were 62 °C, 61 °C, and 61 °C.”
Line 258: “diagram, which”
Line 276: “Analysis of the diagram shows that”
Line 340-341: “indicating that primer No.14 exhibited strong specificity.”
Line 409-410: “Moreover, this technology does not require specialized laboratory equipment, making it faster and easier to operate. ”]

Reviewer 2 Report
Comments and Suggestions for Authors
Dear Authors,
this manuscript is a very interesting and emerging topic of investigation that falls within the scope of the journal. It is focused on the development of useful tools for a rapid diagnosis of HAV in foods. However, some revisions are needed.

A comprehensive review of English language and orthography would be appropriate
Author Response
Response to Reviewer 2 Comments
Thank you very much for your letter and the comments from the referees about the manuscript (Manuscript ID: foods-3473510) entitled “Development of Visual Loop-Mediated Isothermal Amplification Assays for foodborne Hepatitis A Virus" which we submitted to Foods. We have carefully taken all the comments into consideration in preparing our revision, which has resulted in a manuscript that is clearer, more compelling, and broader. We submit here the revised manuscript as well as our point-by-point responses. All changed parts of the revised manuscript were highlighted with blue in the paper.
Thanks very much for your attention to our manuscript. If you have any questions about this paper, please don’t hesitate to let me know.
Point-by-point response to Comments and Suggestions for Authors:
Comments 1: [Line 92-99: Please verify that the kits used and the relative Manufacturer are indicated correctly.]
Response 1: [Thank you for your comments. We have all revised it in correct way. It could be found from Line:93-105.
“HNB dyes (G1218) was purchased from Shanghai Solarbio Biotechnology Co., Ltd. Hepatitis A virus RNA standard (BNCC 369773) was obtained from North China Biological Technology Co., Ltd., Henan, China. Hepatitis A virus inactivation particles was purchased from Tianjin KMD Technology Development Co., Ltd. AgarD, 50×TAE buffer, all primers, and plasmid standards were obtained from Sangon Bioengineering (Shanghai) Co., Ltd. Tiangen Bacterial Genomic DNA Extraction Kit (DP302) was obtained from Tiangen Biochemical Technology (Beijing) Co., Ltd. High Pure dNTPs (10 mM) was obtained from TransGen Biotech Co., Ltd. RNApure Fast Tissue&Cell Kit (CW0599T) and RT-qPCR Mix(CW3371) were obtained from Kangwei Century Biotech Co., Ltd. WarmStart® RTx Reverse Transcriptase (M0380S) was obtained from NEB Biological Technology (Beijing) Co.,Ltd. Lateral flow test strips were obtained from Wuhan Genenode Biotechnology Co., Ltd. Cresol red dye and PEG8000 were from Shanghai McLean Biochemical Technology Co., Ltd.”]
Comments 2: [Line 102-107: The procedure for the recovery of the bacterial culture and DNA extraction could be described in a discursive manner.]
Response 2: [Thank you for your comments. I have revised it in a more discursive way. It could be found from Line:108-123.
“We take out Listeria monocytogenes, E.coil and Vibrio parahaemolyticus from the -80 ℃ refrigerator. This step of bacterial resuscitation is of fundamental importance. According to a 1:100 ratio, 50 μL of the bacterial solution is sequentially added to 5mL of the LB culture solution. Then we placed this mixture in a shaker operating at 220 r/min and incubated at 37 ℃ overnight for 12 hours. After the initial 12 hour incubation, taking the plate of the resuscitated bacterial solution to perform line - drawing. Continue the incubation for another 12 - 18 hours. After checking that there were no stray bacteria, single colonies were picked and transferred to 10mL of LB culture solution once again, and then cultured overnight in the shaker for subsequent experimental operations.”
“Take the 1 ml of three kinds of bacteria after recovery (OD600=1.0 or so). The genomic DNA of Listeria monocytogenes, E.coli and Vibrio parahaemolyticus was extracted according to the instruction of Tiangen Bacterial Genomic DNA Extraction Kit (DP302), and the concentration and quality of the DNA was checked by ultra-micro UV spectrophotometer and stored in the refrigerator at -20 ℃ for further use.”]
Comments 3: [Line 110 – 111: Please indicate the exact amount (volume) of bacteria taken for the extraction of DNA.]
Response 3: [Thanks for your good advice. I have added “1 ml”. It could be found from Line:118.]
Comments 4: [Line 112 – 113: Please, verify if the name “Tigen Bacterial Genomic DNA Extraction Kit” is correct.]
Response 4: [Thank you for your comments. It’s not correct. I have revised it to “Tiangen Bacterial Genomic DNA Extraction Kit (DP302)”. It could be found from Line:120-121.]
Comments 5: [Line 118 - 143: Please, describe the procedure in a detailed and discursive manner, indicating the quantity of digestive gland weighed and used for the RNA extraction. Moreover, indicate the name and the Manufacturer of the kit used for the extraction of RNA.]
Response 5: [Thanks for your good advice. I have revised it in a more detailed and discursive manner, added “Each sample collect 2 g of the digestive gland into a 1.5 ml centrifuge tube” , revised “RNA was extracted immediately according to the RNApure Fast Tissue&Cell Kit (CW0599T) instructions from Kangwei Century Biotech Co., Ltd.” ... It could be found from Line:125-141.
“We wash the surface of the fresh bivalve molluscan shellfish samples, naturally drain the water on the surface of the shell, so that there is no flowing water on the surface, and weigh the weight of the larger and more active bivalve shell samples. Then put it back in a foam tank filled with ice to next step. We wear a mask and put an ice bag on a clean and flat operating table covered with tin paper to maintain a low temperature environment, cut the shell with a surgical razor ,and collect the digestive gland into small pieces. Each sample collect 2 g of the digestive gland into a 1.5 ml centrifuge tube, weigh it and record the data. Gloves and blades were replaced for each sample. According to the method of virus enrichment in ISO 15216–2:2019 [18], some modifications were made [19,20] : After the digestive gland was extracted, we added 500 μL normal saline ,and ground it at low temperature. Then, we take 400 μL of the supernatant and mix it evenly with a 40% PEG 8000 solution (prepared with enzyme-free water) to make the final concentration of PEG 8000 reach 10%. Then centrifuged the mixture at 12,000 rpm for 30 min at 4 ℃. Remove excess supernatant, and the final volume of the mixture was 200 μL. RNA was extracted immediately according to the RNApure Fast Tissue&Cell Kit (CW0599T) instructions from Kangwei Century Biotech Co., Ltd.”]
Comments 6: [Line 163 – 164: Please, indicate the amount of template used and the final volume of the reaction mix.]
Response 6: [ Thanks for your good advice. I have added “25 μL” , “1 μL template for all experiments” in our paper. It could be found from Line:165,174]
Comments 7: [Line 192 – 194: Please, indicate the starting concentration of the Hepatitis A virus and how many dilutions were performed to establish the limit of detection (LOD) in the sensitivity test.]
Response 7: [ Thank you for your insightful comment and kind suggestion. I have revised it to “Different concentrations of 2.59×104 copies/μL, 2.59×103 copies/μL, 2.59×102 copies/μL, 2.59×101 copies/μL, 2.59×100copies/μL RNA were obtained by diluting the 2.59×105 copies/μL hepatitis A virus standard as a 1 μL template to evaluate the detection limit. ”. It could be found from Line:199-201.]
Comments 8: [Line 198 - 199: Please replace the sentence with the following: "The experiment was conducted by three different operators”.]
Response 8: [Thanks for your good advice. I have replaced it. It could be found from Line:206.]
Comments 9: [Line 257 – 258: The sentence “Affect the sensitivity of the reaction, and will not produce a secondary structure between the primers” is not clear. Please, explain in a better way this sentence.]
Response 9: [ Thanks for your good advice. I have revised it to “This indicates that during the LAMP reaction, there will be no mismatch between the primers in the system, resulting in a decrease in the primer concentration in the system and affecting the sensitivity of the reaction,and no secondary structure between the primers will be generated”. It could be found from Line:263-266]
Comments 10: [Figure 3: Please, modify the picture indicating the numbers from 1 to 6 above the wells and the “N” letter above the last well (negative control).]
Response 10: [ Thanks for your good advice. I'm sorry I didn't get your exact meaning. What do you want me to modify Figure 3 into ? Please see PDF]
Comments 11: [Figure 4: The picture of the gel shows 6 wells, while in the legend are described a series of wells from 2 to 8, apart the wells “M” (DNA 100 Marker) and “1” (control group).]
Response 11: [ Thank you for your insightful comment and kind suggestion. I'm missing two. I have revised it to “Figure 4. Specificity test results of the method. M: DNA 100 Marker; 1-7: Sapovirus, Rotavirus, Norovirus, E.coli, HAV, Listeria monocytogenes, Vibrio parahaemolyticus; N: Control group”.It could be found from Line:343-344.
The re-do experiment image is in PDF. I'll up the new images to editor. Thank you very much !]
Comments 12: [Line 352-353: This sentence is poor clear. Please explain it in a better way.]
Response 12: [Thanks for your good advice. I have revised it to “20 set artificial contaminated samples were used as template, 1μL template for RT-LAMP-LFD experiment, 5μL template for RT-qPCR experiment”. It could be found from Line:360-361]
Comments 13: [Line 368: It would be better to indicate figure 8 at the end of the paragraph (line 372).]
Response 13: [ Thank you for your comments. I have revised it to “The positive samples were numbered as 10, 13, 26, 72, 96, 100, 102, 107, 114, 115, 126, 127 and 133 (Figure 8)”. It could be found from Line:381-382 ]
Comments 14: [Finally, a comprehensive review of English and orthography would be appropriate.]
Response 14: [ Thank you for your comments. I wholeheartedly agree with your suggestion. A comprehensive review of English and orthography will surely enhance the quality of the manuscript. I proofread it carefully and corrected some minor errors . Such as, we revised “6 ^ FAM ” to “6^FAM” - Line:24, rephrased “GII Norovirus, Rotavirus, and Sapovirus” - Line:89, revised “group.Both ” to “group. Both” - Line:150-151, revised “BstDNA polymerase” to “Bst DNA polymerase” - Line:166, revised “LAMP-specific” to “LAMP specific” - Line:193, revised “the C,T line” to “the C, T line” - Line:210, revised “Stand at room temperature for 5 minutes and wait for the results. ” to “Leave it at room temperature for 5 minutes and wait for the result.” - Line:314-315, revised “ “Science and Technology Innovation Action Plan” Agricultural Field Project (22N31900600) ” to “Shanghai “Science and Technology Innovation Action Plan” Agricultural Field Project (22N31900600).” - Line:418-419, 421-422]

Round 2
Reviewer 2 Report
Comments and Suggestions for Authors
Dear Authors,
the revisions made to the manuscript ID foods-3473510 have improved it and made it more suitable for publication in Foods. However, I think that paragraphs 2.2 and 2.3 of the Materials and Methods, could be further improved by using an impersonal description (without the use of the pronoun "we"). Regarding Figure 3, it has been improved. However, the legend should follow the order of description from left to right. So, after M = DNA 100 marker, N = negative control, should be cited, followed by wells from 1 to 6. Finally, I suggest a further spelling check.
Author Response
Response to Reviewer 3 Comments
Dear Professor,
Thank you again for all the comments for “Development of Visual Loop-Mediated Isothermal Amplification Assays for foodborne Hepatitis A Virus” (Manuscript ID: foods-3473510). Those comments are all valuable and helpful for revising and improving our paper. We have revised our manuscript according to the comments and have highlighted the revised portion in the paper with green. Our point-by-point responses to the comments are given below; all the page and line numbers refer to those in the revised manuscript.
Thanks very much for your attention to our manuscript. If you have any questions about this paper, please don’t hesitate to let me know.
Point-by-point response to Comments and Suggestions for Authors:
Comments 1: [Dear Authors,
the revisions made to the manuscript ID foods-3473510 have improved it and made it more suitable for publication in Foods. (1) However, I think that paragraphs 2.2 and 2.3 of the Materials and Methods, could be further improved by using an impersonal description (without the use of the pronoun "we"). (2) Regarding Figure 3, it has been improved. However, the legend should follow the order of description from left to right. So, after M = DNA 100 marker, N = negative control, should be cited, followed by wells from 1 to 6. (3) Finally, I suggest a further spelling check.]
Response 1: [
(1)Thank you for your kind suggestion ! I have rephrased paragraphs 2.2 and 2.3 of the Materials and Methods by using an impersonal description (without the use of the pronoun "we"). It could be found from Line:108-119,125-139.
Line 108-119: “Bacterial pure culture[16,17]: Listeria monocytogenes, E. coli and Vibrio parahaemolyticus were taken out from the -80 ℃ refrigerator. This step of bacterial resuscitation is of fundamental importance. According to a 1:100 ratio, 50 μL of the bacterial solution was sequentially added to 5mL of the LB culture solution. The mixture was then placed in a shaker operating at 220 r/min and incubated at 37 ℃ overnight for 12 hours. After the initial 12 hour incubation, the plate of the resuscitated bacterial solution was taken to perform line - drawing. The incubation was continued for another 12 - 18 hours. After checking that there were no stray bacteria, single colonies were picked and transferred to 10mL of LB culture solution once again, and then cultured overnight in the shaker for subsequent experimental operations.
Bacterial DNA extraction: 1 ml of three kinds of bacteria after recovery (OD600=1.0 or so) was taken. The genomic DNA of Listeria monocytogenes, E.coli and Vibrio parahaemolyticus was extracted according to the instruction of Tiangen Bacterial Genomic DNA Extraction Kit (DP302), and the concentration and quality of the DNA was checked by ultra-micro UV spectrophotometer and stored in the refrigerator at -20 ℃ for further use.”
Line 125-139: “The surface of the fresh bivalve molluscan shellfish samples were washed, and the water on the surface of the shell was naturally drained so that there was no flowing water on the surface. The larger and more active bivalve shell samples were weighed. Then they were put back in a foam tank filled with ice for the next step. A mask was worn, and an ice bag was placed on a clean and flat operating table covered with tin paper to maintain a low temperature environment. The shell was cut with a surgical razor, and the digestive gland was collected into small pieces. 2 g of the digestive gland from each sample was collected into a 1.5 ml centrifuge tube, weighed, and the data was recorded. Gloves and blades were replaced for each sample. According to the method of virus enrichment in ISO 15216–2:2019[18], some modifications were made[19,20] : After the digestive gland was extracted, 500 μL normal saline was added, and it was ground at low temperature. Then, 400 μL of the supernatant was taken and mixed it evenly with a 40% PEG 8000solution (prepared with enzyme-free water) to make the final concentration of PEG 8000 reach 10%. The mixture was then centrifuged at 12,000 rpm for 30 min at 4 ℃. Excess supernatant was removed, and the final volume of the mixture was 200 μL. RNA was extracted immediately according to the RNApure Fast Tissue&Cell Kit (CW0599T) instructions from Kangwei Century Biotech Co., Ltd.”
(2)Thank you for your valuable comments ! It's true that my legend doesn't follow the order in which it is presented. I have revised it from “Figure 3. The results of LAMP reaction under different concentrations of RNA. M: DNA 100Marker;1-6 : 1μL RNA template concentration of 2.59×100,2.59×101,2.59×102,2.59×103,2.59×104, 2.59×105 copies/μL; N:negative control.” to “Figure 3. The results of LAMP reaction under different concentrations of RNA. M: DNA 100Marker; N: negative control; 1-6: 1μL RNA template concentration of 2.59×100, 2.59×101, 2.59×102, 2.59×103, 2.59×104, 2.59×105 copies/μL. It could be found from Line:330-332.
Following your suggestion, I have revised the legend for the rest of the article, which also has the same problem.It could be found from:
Line 343-344: “Figure 4. Specificity test results of the method. M: DNA 100Marker; N: Control group; 1-7: Sapovirus, Rotavirus, Norovirus, E.coli, HAV, Listeria monocytogenes, Vibrio parahaemolyticus. ”
Line 354: “Figure 5. Repeated test of LAMP-LFD. M: DNA 100Marker; N: negative control.”
Line 370-371: “Figure 7. Results of artificial pollution simulation experiments. M: DNA 100Marker; +:positive control; -:negative control.”
(3)Thank you for your valuable comments ! Following your suggestion, I have carefully checked the full text , and rephrased the sentences that may be ambiguous, spelling miss, and problematic format.
Line 29: revised “positive detection rate” to “positive detection ratio”.
Line 55-56: revised “The incidence of HAV is significantly higher in areas with raw food habits” to “The incidence of HAV is significantly higher in areas where people have raw food habits”.
Line 57: revised “When HAV infects human liver cells, it replicates itself.” to “When HAV infects human liver cells, it replicates.”. delete “itself”.
Line 88, 108, 344: revised “E.coil” to “E. coil ”.
Line 147: revised “keep them be HAV - negative” to “keep them HAV - negative”. ]
